# Refined Test and Evaluation Method of Side Viewing Angle of LED Display Module

**Shuo Huang** [1,2], **Xifeng Zheng** [1,3,*], **Fengxia Liu** [1,2], **Hui Cao** [3] **and Xinyue Mao** [1,3]

1  Changchun Institute of Optics, Fine Mechanics and Physics, Chinese Academy of Sciences, Changchun 130033, China
2  University of Chinese Academy of Sciences, Beijing 100049, China
3  Changchun Cedar Electronics Technology Co., Ltd., Changchun 130033, China
*  Correspondence: zhengxf@ccxida.com

**Abstract:** In order to classify light-emitting diode (LED) display units with a specific luminous direction and improve the uniformity of the display screen, this paper proposes a classification method for the basic component of the LED display, i.e., the display module. According to the relationship between the light intensity distribution and the viewing angle of the packaged LED, the brightness inflection point was used as the fixed position of the two side array cameras, and the shooting was carried out from the normal direction of the display module and from the viewing angle directions of the left and right sides, respectively. The established classification model of the relative deviation between classes was used to evaluate the luminous consistency of the display module. The LED display module was directly classified according to the side viewing angle. The experiment considered the matching with human vision and controlled the number of product types. The results show that the brightness uniformity of the display module selected by this method increased by 3.31%, which basically ensures that the yield is not reduced due to the inconsistency of the side viewing angle, and this improvement can be judged in advance without building a screen, which not only saves time but also reduces manpower. This method has been stably and feasibly applied to engineering practice and can be classified according to customers' requirements for LED display perspective.

**Keywords:** LED; display; viewing angle; uniformity

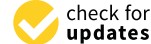



## 1. Introduction

Light-emitting diodes (LEDs) are widely used in lighting and display fields due to their high brightness and low power consumption [1]. The uniformity of LED display products is mainly reflected in three aspects: brightness, chromaticity and viewing angle characteristics. The brightness and chromaticity can be improved using the method of correction. Due to the optical properties of the LED light-emitting tube, each LED light-emitting tube presents different light intensity distributions at multiple angles in space. This problem depends on the structure of the packaged semiconductor chip and the packaging form of the LED [2,3]. Displays with too large a range of luminescence direction specificity cannot be changed even by correction.

The LED large screen is assembled and spliced by the display unit (hereinafter referred to as the display module). Due to the different spatial angle light intensities of the LED light-emitting chip and the difference in the production process, the luminance and color of the display module at different viewing angles are different, resulting in inconsistencies in the display angle of the displayed image [4]. Therefore, the inconsistency of the viewing angle of the display screen has become an important factor affecting the quality of the display screen. Bai Yanyan analyzed the influence of viewing angle on the brightness uniformity of the display screen and came to the conclusion that, the more the viewing angle deviates

from the normal direction, the worse the display screen uniformity is and proposed a display screen uniformity evaluation method based on temperature and viewing angle [5]. Zhu Xin et al. proposed a simple, low-cost method to enhance angular color uniformity of color-mixed light-emitting diodes by introducing a diffuse coating layer [6]. Xia Daxue et al. proposed a full-color viewing angle compensation algorithm for the brightness difference at the splicing of the cabinets, which compensates the RGB three-color image pixel by pixel [7]. The above literature all show that consistency of the display viewing angle has always been a difficult problem to solve in the industry. In this paper, starting from the direction of the spatial distribution of the luminous characteristics of the display unit, theoretical analysis and testing are carried out to evaluate the specificity, so as to screen and classify the display modulesto improve the display uniformity of the display screen.

The display effect is closely related to human visual characteristics [8]. According to Weber's law, human eyes are sensitive to relative luminance deviation rather than to absolute luminance deviation. Too high a background brightness will reduce the human eye's perception of non-uniform defects on the screen [9]. Therefore, in this paper, the relative brightness differences of luminescent units in the range of horizontal viewing angles were evaluated by combining the characteristics of human vision, and the display units were classified and the display brightness uniformity was evaluated based on the influence of the side view on the observers. This study has guiding significance for the production process control of LED display modules.

## 2. Methods

### 2.1. LED Luminous Intensity and Distribution

The luminescence intensity of an LED represents how strongly or weakly it emits light in a certain direction. The different light intensity at different spatial angles directly affects the minimum viewing angle of the LED display device. If the light intensity distribution of the LED super-large color display is inconsistent, the audience facing the display at a large angle will see a color cast image.

Luminescence intensity is an important parameter to characterize light-emitting devices. The unit of luminous intensity is candela (cd), which represents the luminous flux (in lumens) emitted by the light source within a unit solid angle of a given direction [10].

The angular distribution of the luminous intensity $I(\theta)$ describes the luminous intensity distribution of the LED light-emitting device in all directions in space. The relative luminescence intensity along the normal direction is defined as 1, and the larger the angle away from the normal direction, the smaller the relative luminescence intensity is [11].

Assuming that all LED devices on the LED module are point light sources with the same luminous intensity distribution and luminous flux, the approximate Lambertian light source, that is, the luminous intensity distribution of the LED device can be expressed by the power function of the viewing angle cosine [12], which is given by Equation (1):

$$I(\theta) = I_0 \cos^m(\theta) \tag{1}$$

Luminance refers to the ratio of the luminous intensity of the illuminant to the area of the light source, that is, the luminous intensity per unit projected area. The human eye can intuitively feel the luminance, so this paper judges the viewing angle characteristics by collecting the luminance. Then, the luminance $L$ (cd/m$^2$) of a point P on the target plane at a distance from the LED light source $r$ is as follows:

$$L(r, \theta) = \frac{I_0 \cos^m(\theta)}{r^2} \tag{2}$$

Among them, $I_0$ is the normal luminous intensity of the LED, $\theta$ is the luminous angle of the LED and $m$ is the light distribution curve of the LED light-emitting chip, which is related to the packaging material of the LED package [13].

In the practical engineering application of Mini LED flip-chip LED display, the light-emitting chip needs to be fixed on a substrate, through a process called "bonding", and in this process, the optical axes of all LED light-emitting chips are required to be parallel [14], but due to differences in technology, the optical axis of one or some LED light-emitting chips is offset, as shown in Figure 1.

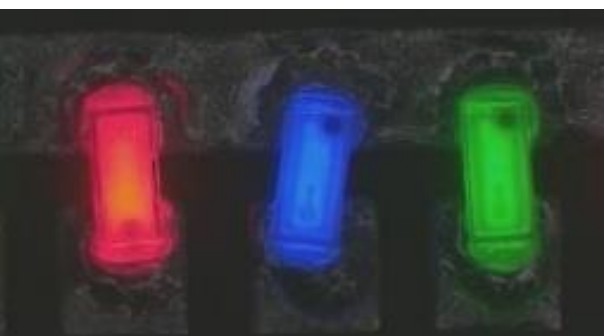

**Figure 1.** The optical axis of the LED light-emitting chip is offset during bonding (red, blue and green from left to right).

The bonding causes the optical axis to shift, resulting in a change in the maximum luminous intensity at the angle. Let the spatial horizontal angle offset be $\Delta\theta$ (the vertical angle is not common in engineering, so this article only studies the horizontal angle), then the luminance changes to

$$L(r, \theta + \Delta\theta) = \frac{I_0 \cos^m(\theta + \Delta\theta)}{r^2} \tag{3}$$

Since the distance $r$ of the optical axis offset of the light-emitting chip relative to a certain point in space is extremely small, it is considered that $r$ does not change. Divide Equation (3) by Equation (2) to get the change ratio of the luminance after the optical axis shift:

$$\frac{L(r, \theta + \Delta\theta)}{L(r, \theta)} = (cos\Delta\theta - tan\theta sin\Delta\theta)^m \tag{4}$$

After the bonding is finished, the light-emitting chip needs to be encapsulated to protect the light-emitting chip. During this process, the particulate matter and thickness of the adhesive layer will affect the light-emitting viewing angle and light-emitting intensity of the light-emitting chip. Assuming that the horizontal angle offset of the light-emitting chip is $\Delta\theta'$, and the luminous intensity loss during the packaging process is $\Delta I$, the luminance change is as follows:

$$\frac{L(r, \theta + \Delta\theta + \Delta\theta')}{L(r, \theta)} = \left(\cos(\Delta\theta + \Delta\theta') - tan\theta \sin(\Delta\theta + \Delta\theta')\right)^m - \frac{\Delta I}{I_0 \cos^m(\theta)} \tag{5}$$

### 2.2. Relationship between Brightness and Angle of the Display Module

The brightness after packaging is the real brightness of the LED display. Due to the random and unmeasurable horizontal angle offset caused by the bonding and packaging process, the brightness variation is unknown.

Therefore, the LED display module packaged by COB (chips on board) is measured by using the equipment in Figure 2. The measurement must meet the following requirements: the axis of the luminance meter passes through the rotation axis of the turntable, luminance meter lens' center height from the ground is the same as the module's center height from the ground, the rotation axis of the turntable and the surface of the display module coincide and the center line of the module coincides with the rotation axis of the turntable. The display module is rotated with a step of 9 degrees, and the brightness at the center of the

display module at each angle is measured to explore the change of the luminance with the angle.

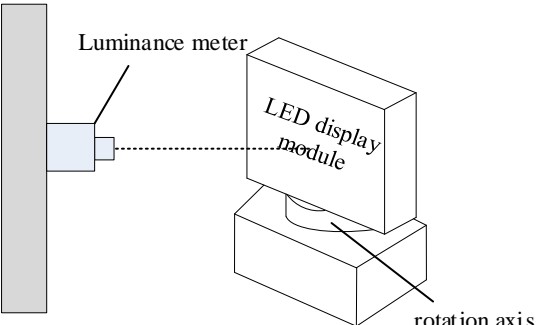

**Figure 2.** Visual angle measuring equipment.

The attenuation curve of the luminance of the three primary colors in the display module along with the angle is shown in Figure 3, which indicates the relationship between viewing angles and normalized luminance.

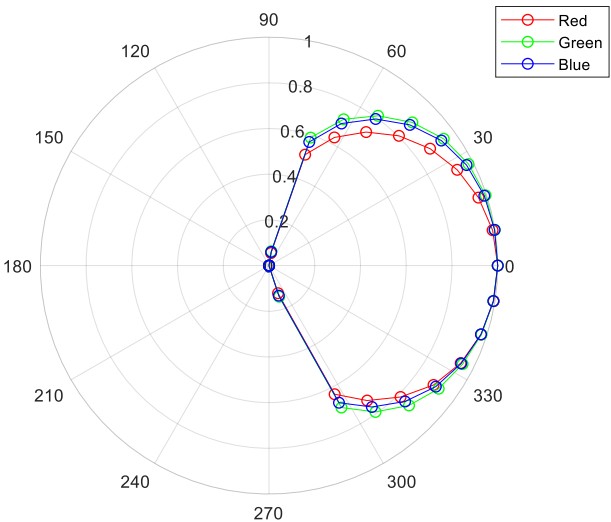

**Figure 3.** The attenuation curve of the luminance of the three primary colors of the display module.

Usually, the measurement of the viewing angle uses the device shown in Figure 2, but this device measures the entire light field distribution [15], which takes a long time and is complicated to operate, and cannot select display modules with inconsistent luminous viewing angles in real time among a large number of display modules. In view of the above problems, this paper explores whether it is possible to find a fixed angle to directly select the display modules with inconsistent light emission, which not only saves time and simplifies the operation but also lays a foundation for engineering applications.

Since the measured data are scattered points, the luminance attenuation curve is fitted in Cartesian coordinate system. The eighth-order Gaussian function fitting in the fitting method is closest to the true value. The fitting curve is shown in Figure 4.

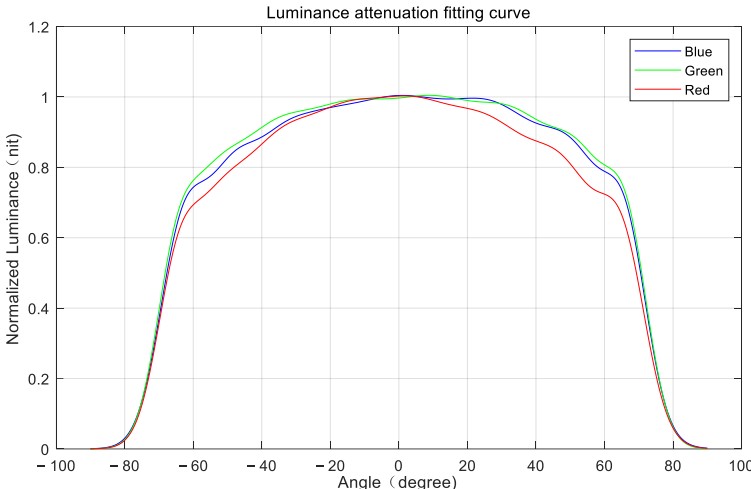

**Figure 4.** Luminance attenuation fitting curve of the three primary colors.

The inflection point of the curve was then found. Under the inflection point, the brightness of the display module did not change significantly, so it was impossible to distinguish whether the luminescence angle of the display module was consistent. The luminance of the display module at the inflection point is different. Taking this point as a fixed angle measurement, the module with inconsistent luminance can be picked out without measuring the overall luminance distribution so as to save time and simplify the operation. We call this point the luminance inflection point.

The luminance inflection point of the above curve is the second-order extreme point of the curve. The quadratic derivation of the luminance attenuation fitting curves of the three primary colors is shown in Figure 5. Taking 0° as the center, the minimum point of the second derivative is the first luminance inflection point, and the maximum point of the second derivative is the second luminance inflection point. However, the serious attenuation of the brightness at this time is no longer suitable for viewing angle. Therefore, in this experiment, the first luminance inflection point, namely the second-order derivative minimum of the luminance luminescence fitting curve, was selected as the experimental object to explore the improvement of side view uniformity.

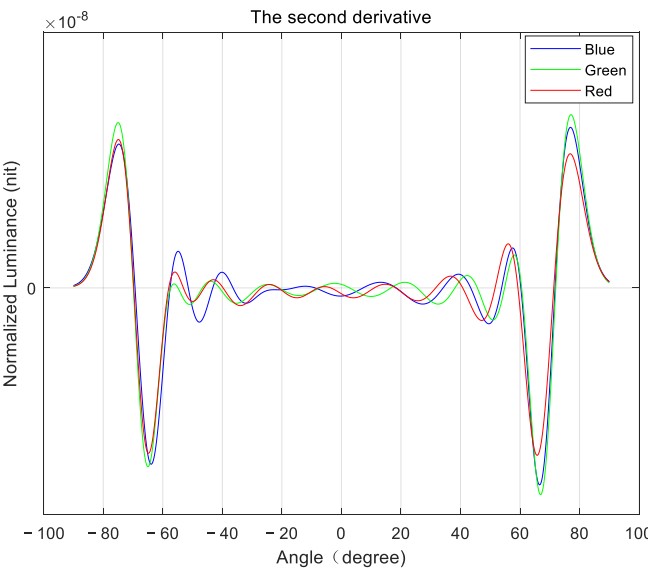

**Figure 5.** The second derivative of the luminance attenuation fitting curve.

*2.3. Classification Model with Equal Relative Deviation between Classes*

The main purpose of this paper is to select the display module with the same luminous angle of view among a large number of display modules to improve the display effect of the whole screen. In view of this feature and combined with statistical law, this paper proposes a classification model of equal relative deviation between categories, which divides the display modules into *n* classes. If the relative deviation between classes is equal, then the display modules of the same class have the same luminous viewing angle so as to improve the display effect of the whole display screen.

The brightness in the normal direction of the display module is represented by $L_i(n)$, and the brightness when the angle is $\theta$ is represented by $L_i(\theta)$, where *i* is the serial number of the display module. The brightness deviation value $x_i(\theta)$ represents the ratio of the brightness at the angle $\theta$ to the brightness in the normal direction. The calculation method is as follows:

$$x_i(\theta) = \frac{L_i(\theta)}{L_i(n)} \tag{6}$$

The classification feature value is used to determine the classification interval of the display module. If five categories are used as an example, the calculation method of the classification feature value is as follows:

$$\frac{X_0 - X_1}{(X_0 + X_1)/2} = \frac{X_1 - X_2}{(X_1 + X_2)/2} = \frac{X_2 - X_3}{(X_2 + X_3)/2} = \frac{X_3 - X_4}{(X_3 + X_4)/2} = \frac{X_4 - X_5}{(X_4 + X_5)/2} \tag{7}$$

where $X_0 = max(x_i(\theta))$, $X_5 = min(x_i(\theta))$, that is, $X_0$ and $X_5$ are the maximum and minimum values of the luminance deviation value, respectively.

The classification eigenvalues $X_1, X_2, X_3, X_4$ can be obtained by solving the equations of the continuous equation:

$$\begin{cases} X_1 = \sqrt[5]{(X_5 \cdot X_0^4)} \\ X_2 = \frac{X_1^2}{X_0} \\ X_3 = \frac{X_1^3}{X_0^2} \\ X_4 = \frac{X_1^4}{X_0^3} \end{cases} \tag{8}$$

The brightness deviation value is compared with the classification feature value to determine the category of the display module: [0] means the brightness is normal, [1] means the brightness is slightly bright, [2] means the brightness is brighter, [−1] means the brightness is slightly bright dark class, [−2] means darker class.

$$\begin{cases} x_i(\theta) < X_1 & \in [2] \\ X_1 < x_i(\theta) < X_2 & \in [1] \\ X_2 < x_i(\theta) < X_3 & \in [0] \\ X_3 < x_i(\theta) < X_4 & \in [-1] \\ x_i(\theta) > X_4 & \in [-2] \end{cases} \tag{9}$$

The same type of display modules have the same luminous viewing angle and the same perceptual brightness of the human eye. The same type of display modules can be built into the screen, which can significantly improve the uniform effect of the display screen. The difference of the luminescence angle of the adjacent module is acceptable to the human eye, while the difference of the luminescence angle of the cross-class is not acceptable to the human eye.

The luminescence view is closely related to the packaging process. If the packaging process of the module is the same, the same classification eigenvalue can be used. If the process is changed, the classification eigenvalue needs to be redetermined. The above steps must then be repeated to re-determine the classification feature values.

## 3. Experiments

The main process of the experiment is as follows: use the equipment in Figure 2 to measure the brightness change of the LED display screen from $-90°$ to $+90°$ in the horizontal direction, select a suitable angle to fix the camera by the method in Section 2.2 and build an experimental platform. In this paper, the display module with the pixel spacing of 1.19 mm was used as a sample for the experiment. First, the calibration of the camera was completed to determine the camera calibration coefficient, then the classification feature values $X_1$, $X_2$, $X_3$, $X_4$ were determined, and finally, this device was used to select modules with consistent lighting angles, and this experimental method was used to select the display screen arbitrarily built in the sample. The brightness uniformity of the selected display screens was then compared to obtain the experimental results.

The camera selected for the experiment was a CMOS camera ( model: acA3800-10gm, BASLER, Germany ) with a focal length of 12 mm and a resolution of $2748 \times 3840$; the luminance meter was a Konica Minolta CS-2000 spectroradiometer, the applicable wavelength range was 380~780 nm. The lowest brightness it can measure is 0.003 cd/m$^2$. The highest reachable measurement contrast was 100,000:1. The operating temperature range of the instrument was 5 °C ~35 °C, which was similar to the actual working environment. Using this instrument, the luminance, color coordinates, color temperature and other information of the display screen can be easily obtained, which meets the needs of this experiment.

### 3.1. Determination of the Position of the Side-View Camera

The fixed position of the camera can be obtained by the second derivative of the luminance attenuation fitting curve in Figure 5. The first brightness inflection point appears at $\theta = -65°$ and $\theta = 69°$, so this experiment selected the fixed angles $\theta = -65°$ and $\theta = 69°$ as an experimental object to explore the improvement of lateral viewing angle uniformity.

Two cameras were placed at angles of $-65°$ and $69°$ from the normal, and one camera was placed in the normal direction of the display screen. The experimental equipment is shown in Figure 6. The display screen does not need to be rotated, and the time required is short. A large number of modules can be measured, and the display modules with inconsistent viewing angles can be quickly determined.

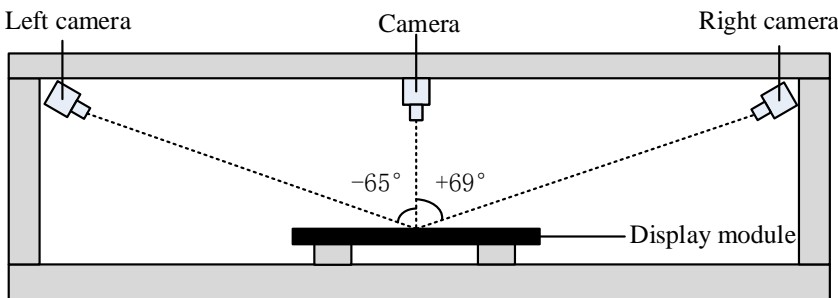

**Figure 6.** Experimental equipment.

### 3.2. Camera Calibration

Since the camera's measurement data are relative values, if multiple cameras measure together, there will be errors between different cameras. Therefore, it is necessary to use the luminance value of the display screen module measured by the luminance meter of the experimental equipment (1) to calibrate the data collected by the camera. After calibration, the measured value of the camera can be considered as the absolute value (under the condition of constant camera position, aperture, focal length and exposure time).

The first step is to take the consistent viewing angle module in the above sample 1. Under the condition of a fixed viewing angle of the same screen and the same module, the three left, middle and right area scan cameras of the experimental equipment (2) should be used to take three photos respectively, the pixel point in the center of the module should be

taken as the coordinate center of the intercepted area and intercept the area of the same size as the window, as shown in the Figures 7–9.

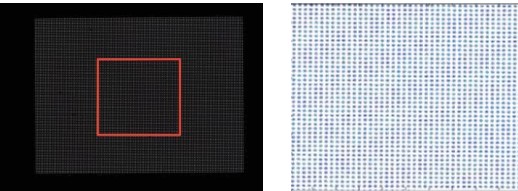

**Figure 7.** Pictures of center camera (Red box in the figure is the window).

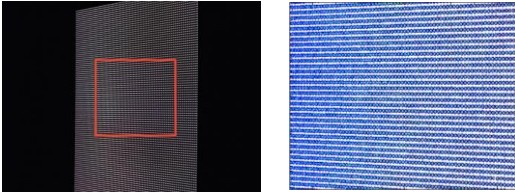

**Figure 8.** Pictures of right camera (Red box in the figure is the window).

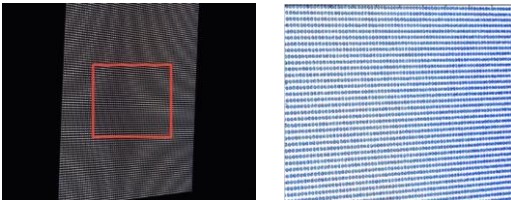

**Figure 9.** Pictures of left camera (Red box in the figure is the window).

The average brightness of the image in this window was extracted, represented by $Gray_{color}(\theta)$. The calibration coefficient of the left and right cameras was calculated, the calibration coefficient of the left camera was $Calibration_{color}(l)$ and the calibration coefficient of the right camera was $Calibration_{color}(r)$, then the calculation method was as follows:

$$Calibration_{color}(l) = \frac{\frac{(Gray_{color}(l) - noise)}{(Gray_{color}(n) - noise)}}{x_{p,color(l)}} \tag{10}$$

$$Calibration_{color}(r) = \frac{\frac{(Gray_{color}(r) - noise)}{(Gray_{color}(n) - noise)}}{x_{p,color(r)}} \tag{11}$$

where $l$ represents $\theta = -65°$, $r$ represents $\theta = 69°$, $n$ represents the normal direction, and *noise* represents the background noise value. Thus, the calibration coefficients of the three cameras were determined, and the camera calibration coefficients were suitable for display modules with different point spacings.

### 3.3. Determination of the Position of the Side-View Camera

Evaluation and classification of inconsistent modules of lateral perspective.

A certain number of display modules with the same pixel spacing were arbitrarily selected as samples, and the experimental equipment in Figure 2 was used to collect the three primary colors of the display module of the left, middle and right cameras, using the same size of the window as the camera calibration in Section 3.2. The mean value of brightness in the window was expressed as $Gray'_{color}(\theta)$. The brightness deviation value

on the left was represented by $x'_{i,color(l)}$, and the brightness deviation value on the right side was represented by $x'_{i,color(r)}$, then:

$$x'_{i,color}(l) = \frac{\left(Gray'_{color}(l) - noise\right)}{\left(Gray'_{color}(n) - noise\right)} / Calibration_{color}(l) \tag{12}$$

$$x'_{i,color}(r) = \frac{\left(Gray'_{color}(r) - noise\right)}{\left(Gray'_{color}(n) - noise\right)} / Calibration_{color}(r) \tag{13}$$

$$x'_i(l) = x'_{i,R}(l) + x'_{i,G}(l) + x'_{i,B}(l) \tag{14}$$

$$x'_i(r) = x'_{i,R}(r) + x'_{i,G}(r) + x'_{i,B}(r) \tag{15}$$

The sum of the calibrated red, green and blue brightness deviation values is called the final brightness deviation value. The final brightness deviation value was substituted on the left $x'_i(l)$ and on the right $x'_i(r)$ into Equation (9) to obtain the left classification feature value of the device and the right-side categorical eigenvalues.

## 4. Discussion

The sample was 171 display modules with a dot pitch of 1.19 mm. The above method was used to screen the display modules with inconsistent viewing angles. The distribution histogram of the left and right classifications is shown in the Figures 10 and 11. Among them, [0] had the largest number of display modules in the normal brightness category, which is in line with expectations.

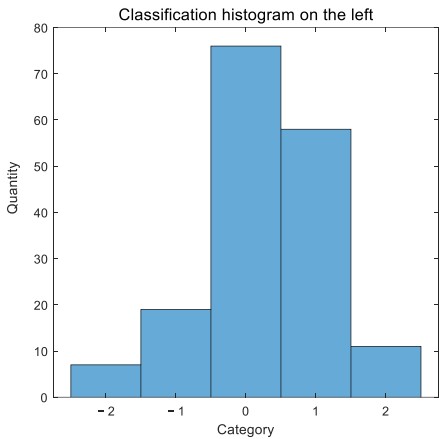

**Figure 10.** Classification histogram on the left.

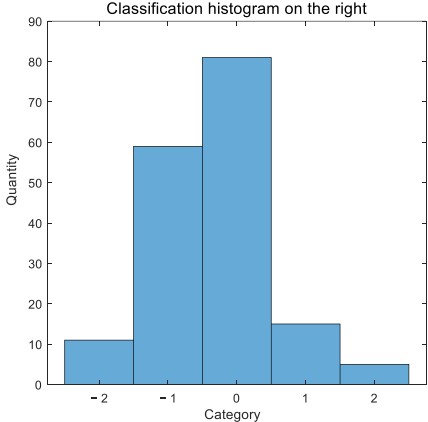

**Figure 11.** Classification histogram on the right.

In order to verify the accuracy of the experimental results, 12 modules in the display module of class [0] were built into a box, as shown in Figure 12, which consisted of 4 columns and 3 rows. The CS2000 luminance meter was the same height as the measured point and the luminance of the display module was measured at a fixed angle $\theta = 69°$, the brightness data is shown in Table 1. At the same time, 12 random display modules were built, and the scheme was also used for measurement, the brightness data is shown in Table 2. (The same method was used for measurement when $\theta = -65°$, which will not be repeated in this paper).

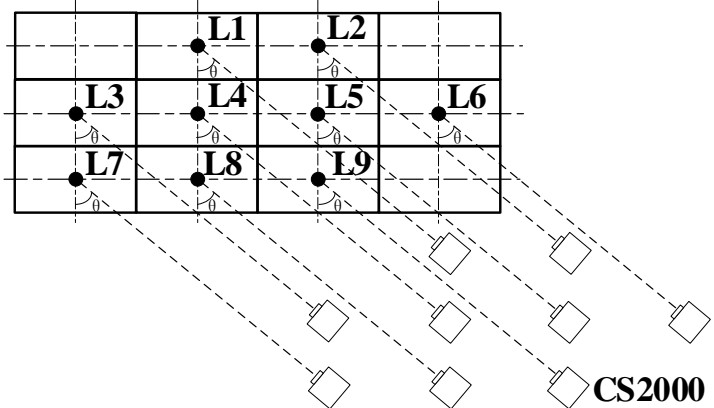

**Figure 12.** Schematic diagram of construction method.

**Table 1.** The brightness of the display module with the same viewing angle selected by this experimental method (cd/m$^2$).

| Position | Red | Green | Blue |
|:---:|:---:|:---:|:---:|
| $L1$ | 332.52 | 593.39 | 150.83 |
| $L2$ | 322.34 | 596.37 | 153.95 |
| $L3$ | 325.33 | 592.73 | 152.69 |
| $L4$ | 323.83 | 588.99 | 149.18 |
| $L5$ | 326.72 | 599.35 | 154.26 |
| $L6$ | 336.52 | 601.4 | 155.94 |
| $L7$ | 345.87 | 595.25 | 153.99 |
| $L8$ | 332.88 | 601.39 | 153.4 |
| $L9$ | 335.5 | 602.17 | 155.93 |
| $L_{MJ}$ | | 98.87% | |

**Table 2.** The brightness of the randomly constructed display module (cd/m$^2$).

| Position | Red | Green | Blue |
|:---:|:---:|:---:|:---:|
| $L1$ | 323.87 | 577.23 | 148.57 |
| $L2$ | 298.97 | 547.66 | 141.5 |
| $L3$ | 311.69 | 579.19 | 149.61 |
| $L4$ | 324.35 | 579.22 | 147.09 |
| $L5$ | 333.19 | 584.3 | 149.89 |
| $L6$ | 323.88 | 590.46 | 151.54 |
| $L7$ | 318.38 | 567.79 | 144.81 |
| $L8$ | 310.65 | 583.98 | 148.78 |
| $L9$ | 319.41 | 608.89 | 150.95 |
| $L_{MJ}$ | | 95.56% | |

According to the calculation method of the brightness uniformity of the display module in "SJ/T 11281-2017Measure methods of light emitting diode(LED)panels" [16], the

brightness uniformity of the display module with consistent side viewing angle selected by this method was 98.87%. The brightness uniformity of the arbitrarily constructed display module was 95.56%, which proves that the experimental method can improve the brightness uniformity of the display screen by 3.31%.

## 5. Conclusions

In this paper, a classification method for viewing angles of an LED display screen is proposed. According to the relationship between the light intensity distribution of the packaged LED and the viewing angle and the classification model of the relative deviation between classes, the luminous consistency of the display module is evaluated. Considering the matching with human vision and the yield of the product, the LED display module is divided into five categories, and the brightness in the same category is consistent. The brightness uniformity of the selected display module increased by 3.31%. This method is based on area array camera acquisition, to get rid of the dependence of the human eye observation, it can effectively improve the brightness uniformity of the side view of the display screen without the need to build a screen or to determine the identification in advance, which saves time and reduces labor costs. This method has been applied to engineering practice and is stable and feasible.

On the basis of this experiment, the module can continue to be classified in more detail. In the process of setting up the screen, it is arranged in turn according to the category, which can increase the product yield and improve the brightness uniformity of the side view.

**Author Contributions:** Conceptualization, S.H. and X.Z.; methodology, S.H. and F.L.; software, F.L.; validation, H.C.; formal analysis, X.M.; investigation, S.H.; resources, X.M.; data curation, F.L.; writing—original draft preparation, S.H.; writing—review and editing, S.H.; visualization, H.C.; supervision, X.Z.; project administration, H.C. All authors have read and agreed to the published version of the manuscript.

**Funding:** This work was funded by the major science and technology special projects of the Jilin Province Science and Technology Development Program of China, grant number 20210301002GX.

**Institutional Review Board Statement:** Not applicable.

**Informed Consent Statement:** Not applicable.

**Data Availability Statement:** Not applicable.

**Acknowledgments:** The authors thank Ding Tiefu for his guidance and advice.

**Conflicts of Interest:** The authors declare no conflict of interest. The funders had no role in the design of the study; in the collection, analyses, or interpretation of data; in the writing of the manuscript; or in the decision to publish the results.

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
