# Peer review of "Refined Test and Evaluation Method of Side Viewing Angle of LED Display Module"

_applsci, doi:10.3390/app12189177_

Round 1
Reviewer 1 Report
The authors approach in the article a method of testing and evaluating the lateral viewing angle of the LED display module. In my opinion, the fort point of the manuscript is the comparison with the ability of visualize of the human eyes.
The paper is suitable for this journal, but in my opinion, it must be improved in some parts of the study before accepted.
1. The introduction could be more explicit and respectively supplemented with bibliographic references in the field of medical eyes and LED display.
2. The link between the part of the text where the mathematical aspects are referred not is highlighted in relation to the numerical value of the equation. (An example equation 2 has no explicit reference in the text above it).
3. The text of Line 159-161 should be reformulated and correlated with both the formula referred in Line (7) and grammatical aspects. The same aspects Line 182 to 185.
4. Given the required model of writing such a study, the last two parts must be completed and corrected. ( Author Contributions, Funding, Data Availability Statement, Acknowledgments, Conflicts of Interest, References).
Reviewer 2 Report
It is an interesting paper.
I have minor suggestions:
1) The authors should better explain the optical details of the luminance meters;
2) there are several misprints. "0.0005cd/m2" should be "0.0005 cd/m^2". I recommend more accuracy in reporting values.
